# Code as Policies: Language Model Programs for Embodied Control

Jacky Liang*[†], Wenlong Huang*, Fei Xia*, Peng Xu*, Karol Hausman*, Brian Ichter*, Pete Florence*, Andy Zeng*

*Robotics at Google

*Abstract*— **Large language models (LLMs) trained on code-completion have been shown to be capable of synthesizing simple Python programs from docstrings [1]. We find that these code-writing LLMs can be re-purposed to write robot policy code, given natural language commands. Specifically, policy code can express functions or feedback loops that process perception outputs (e.g., from object detectors [2], [3]) and parameterize control primitive APIs. When provided as input several example language commands (formatted as comments) followed by corresponding policy code (via few-shot prompting), LLMs can take in new commands and autonomously re-compose API calls to generate new policy code respectively. By chaining classic logic structures and referencing third-party libraries (e.g., NumPy, Shapely) to perform arithmetic, LLMs used in this way can write robot policies that (i) exhibit spatial-geometric reasoning, (ii) generalize to new instructions, and (iii) prescribe precise values (e.g., velocities) to ambiguous descriptions ("faster") depending on context (i.e., behavioral commonsense). This paper presents *code as policies*: a robot-centric formalization of language model generated programs (LMPs) that can represent reactive policies (e.g., impedance controllers), as well as waypoint-based policies (vision-based pick and place, trajectory-based control), demonstrated across multiple real robot platforms. Central to our approach is prompting hierarchical code-gen (recursively defining undefined functions), which can write more complex code and also improves state-of-the-art to solve 39.8% of problems on the HumanEval [1] benchmark. Code and videos are available at https://code-as-policies.github.io**

## I. INTRODUCTION

Robots that use language need it to be grounded (or situated) to reference the physical world and bridge connections between words, percepts, and actions [4]. Classic methods ground language using lexical analysis to extract semantic representations that inform policies [5]–[7], but they often struggle to handle unseen instructions. More recent methods learn the grounding end-to-end (language to action) [8]–[10], but they require copious amounts of training data, which can be expensive to obtain on real robots.

Meanwhile, recent progress in natural language processing shows that large language models (LLMs) pretrained on Internet-scale data [11]–[13] exhibit out-of-the-box capabilities [14]–[16] that can be applied to language-using robots e.g., planning a sequence of steps from natural language instructions [16]–[18] without additional model finetuning. These steps can be grounded in real robot affordances from value functions among a fixed set of skills i.e., policies pretrained with behavior cloning or reinforcement learning [19]–[21]. While promising, this abstraction prevents the LLMs from directly influencing the perception-action feedback loop, making it difficult to ground language in ways that (i) generalize modes of feedback that share percepts and actions e.g., from "put the apple down on the orange" to "put the apple down *when you see* the orange", (ii) express commonsense priors in control e.g., "move *faster*", "push *harder*", or (iii) comprehend spatial relationships "move the apple *a bit to the left*". As a result,

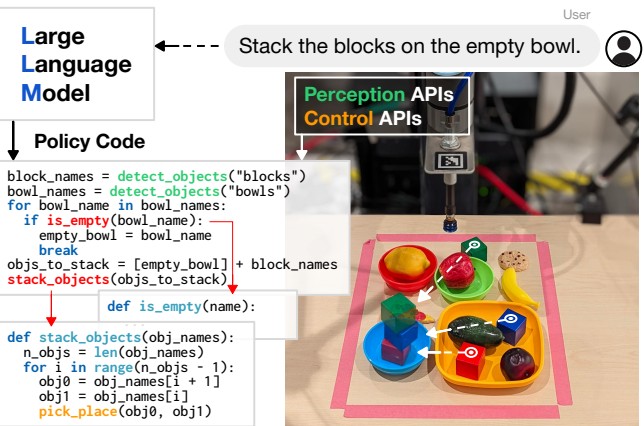

Fig. 1: Given examples (via few-shot prompting), robots can use code-writing large language models (LLMs) to translate natural language commands into robot policy code which process perception outputs, parameterize control primitives, recursively generate code for undefined functions, and generalize to new tasks.

incorporating each new skill (and mode of grounding) requires additional data and retraining – ergo the data burden persists, albeit passed to skill acquisition. This leads us to ask: how can LLMs be applied beyond just planning a sequence of skills?

Herein, we find that *code-writing* LLMs [1], [11], [22] are proficient at going further: orchestrating planning, policy logic, and control. LLMs trained on code-completion have shown to be capable of synthesizing Python programs from docstrings. We find that these models can be re-purposed to write robot policy code, given natural language commands (formatted as comments). Policy code can express functions or feedback loops that process perception outputs (e.g., open vocabulary object detectors [2], [3]) and parameterize control primitive APIs (see Fig. 1). When provided with several example language commands followed by corresponding policy code (via few-shot prompting, in gray), LLMs can take in new commands (in green) and autonomously re-compose the API calls to generate new policy code (highlighted) respectively:

```
# if you see an orange, move backwards.
if detect_object("orange"):
    robot.set_velocity(x=-0.1, y=0, z=0)
# move rightwards until you see the apple.
while not detect_object("apple"):
    robot.set_velocity(x=0, y=0.1, z=0)
```

Code-writing models can express a variety of arithmetic operations as well as feedback loops grounded in language. They not only generalize to new instructions, but having been trained on billions of lines of code and comments, can also prescribe precise values (e.g., velocities) to ambiguous descriptions ("faster" and "to the left") depending on context – to elicit behavioral commonsense:

```
# do it again but faster, to the left, and with a banana.
while not detect_object("banana"):
    robot.set_velocity(x=0, y=-0.2, z=0)
```

Representing code as policies inherits a number of benefits from LLMs: not only the capacity to interpret natural language, but also

---

† Work done while interning at Google.

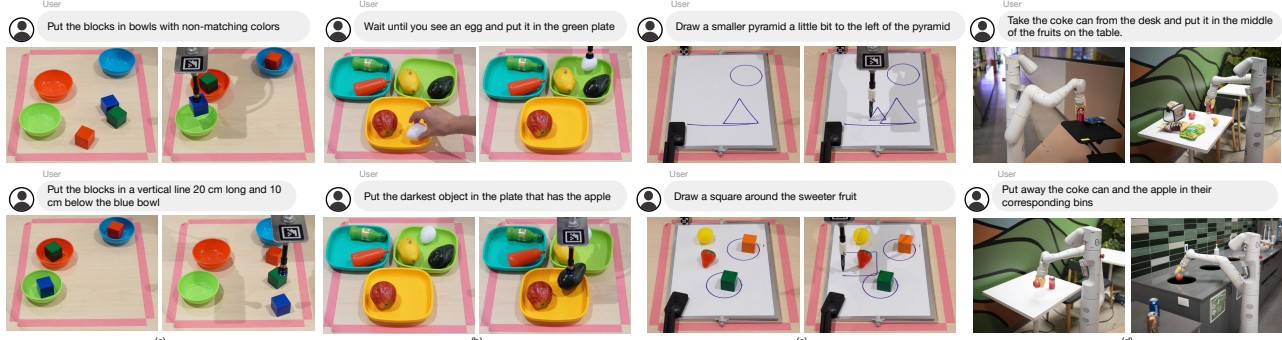

Fig. 2: Code as Policies can follow natural language instructions across diverse domains and robots: table-top manipulation (a)-(b), 2D shape drawing (c), and mobile manipulation in a kitchen with robots from Everyday Robots (d). Our approach enables robots to perform spatial-geometric reasoning, parse object relationships, and form multi-step behaviors using off-the-shelf models and few-shot prompting with no additional training. See full videos and more tasks at code-as-policies.github.io

the ability to engage in human-robot dialogue and Q&A simply by using "say(text)" as an available action primitive API:

```
# tell me why you stopped moving.
robot.say("I stopped moving because I saw a banana.")
```

We present **Code as Policies** (CaP): a robot-centric formalization of language model generated programs (LMPs) executed on real systems. Pythonic LMPs can express complex policies using:

- Classic logic structures e.g., sequences, selection (if/else), and loops (for/while) to assemble new behaviors at runtime.
- Third-party libraries to interpolate points (NumPy), analyze and generate shapes (Shapely) for spatial-geometric reasoning, etc.

LMPs can be *hierarchical*: prompted to recursively define new functions, accumulate their own libraries over time, and self-architect a dynamic codebase. We demonstrate across several robot systems that LLMs can autonomously interpret language commands to generate LMPs that represent reactive low-level policies (e.g., PD or impedance controllers), and waypoint-based policies (e.g., for vision-based pick and place, or trajectory-based control).

Our main contributions are: (i) code as policies: a formalization of using LLMs to write robot code, (ii) a method for hierarchical code-gen that improves state-of-the-art on both robotics and standard code-gen problems with 39.8% P@1 on HumanEval [1], (iii) a new benchmark to evaluate future language models on robotics code-gen problems, and (iv) ablations that analyze how CaP improves metrics of generalization [23] and that it abides by scaling laws – larger models perform better. Code as policies presents a new approach to linking words, percepts, and actions; enabling applications in human-robot interaction, but is not without limitations. We discuss these in Sec. V. Full prompts and generated outputs are in the Appendix, which can be found along with additional results, videos, and code at code-as-policies.github.io

## II. RELATED WORK

**Controlling robots via language** has a long history, including early demonstrations of human-robot interaction through lexical parsing of natural language [5]. Language serves not only as an interface for non-experts to interact with robots [24], [25], but also as a means to compositionally scale generalization to new tasks [9], [17]. The literature is vast (we refer to Tellex et al. [4] and Luketina et al. [26] for comprehensive surveys), but recent works fall broadly into the categories of high-level interpretation (e.g., semantic parsing [25], [27]–[32]), planning [14], [17], [18], and low-level

policies (e.g., model-based [33]–[35], imitation learning [8], [9], [36], [37], or reinforcement learning [38]–[42]). In contrast, our work focuses on the code generation aspect of LLMs and use the generated procedures as an expressive way to control the robot.

**Large language models** exhibit impressive zero-shot reasoning capabilities: from planning [14] to writing math programs [43]; from solving science problems [44] to using trained verifiers [45] for math word problems. These can be improved with prompting methods such as Least-to-Most [46], Think-Step-by-Step [15] or Chain-of-Thought [47]. Most closely related to this paper are works that use LLM capabilities for robot agents without additional model training. For example, Huang et al. decompose natural language commands into sequences of executable actions by text completion and semantic translation [14], while SayCan [17] generates feasible plans for robots by jointly decoding an LLM weighted by skill affordances [20] from value functions. Inner Monologue [18] expands LLM planning by incorporating outputs from success detectors or other visual language models and uses their feedback to re-plan. Socratic Models [16] uses visual language models to substitute perceptual information (in teal) into the language prompts that generate plans, and it uses language-conditioned policies e.g., for grasping [36]. The following example illustrates the qualitative differences between our approach versus the aforementioned prior works. When tasked to "move the coke can a bit to the right":

```
LLM Plan [14], [17], [18]
1. Pick up coke can
2. Move a bit right
3. Place coke can
```
```
Socratic Models Plan [16]
objects = [coke can]
1. robot.grasp(coke can)  open vocab
2. robot.place_a_bit_right()
```

plans generated by prior works assume there exists a skill that allows the robot to move an object a bit right. Our approach differs in that it uses an LLM to directly generate policy code (plans nested within) to run on the robot and avoids the requirement of having predefined policies to map every step in the plan:

```
Code as Policies (ours)
while not obj_in_gripper("coke can"):
    robot.move_gripper_to("coke can")
robot.close_gripper()
pos = robot.gripper.position
robot.move_gripper(pos.x, pos.y+0.1, pos.z)
robot.open_gripper()
```

Our approach (CaP) not only leverages logic structures to specify feedback loops, but it also parameterizes (and write parts of) low-level control primitives. CaP alleviates the need to collect data and train a fixed set of predefined skills or language-conditioned policies – which are expensive and often remain domain-specific.

**Code generation** has been explored with LLMs [1], [48] and without [49]. Program synthesis has been demonstrated to be capable of drawing simple figures [50] and generating policies that solve 2D tasks [51]. We expand on these works, showing that (i) code-writing LLMs enable novel reasoning capabilities (e.g., encoding spatial relationships by leaning on familiarity of third party libraries) without additional training needed in prior works [35], [36], [52]–[56], and (ii) hierarchical code-writing (inspired by recursive summarization [57]) improves state-of-the-art code generation. We also present a new robotics-themed code-gen benchmark to evaluate future language models in the robotics domain.

## III. METHOD

In this section, we characterize the extent to which pretrained LLMs can be prompted to generate code as policies – represented as a set of language model programs (LMPs). Broadly, we use the term LMP to refer to any program generated by a language model and executed on a system. This work investigates Code as Policies, a class of LMPs that maps from language instructions to code snippets that (i) react to perceptual inputs (i.e., from sensors or modules on top of sensors), (ii) parameterize control primitive APIs, and (iii) are directly compiled and executed on a robot, for example:

```
# stack the blocks in the empty bowl.
empty_bowl_name = parse_obj('empty bowl')
block_names = parse_obj('blocks')
obj_names = [empty_bowl_name] + block_names
stack_objs_in_order(obj_names=obj_names)
```

Input instructions are formatted as comments (green), which can be provided by humans or written by another LMP. Predicted outputs from the LLM (highlighted) are expected to be valid Python code, generated autoregressively [11], [12]. LMPs are few-shot prompted with examples to generate different subprograms that may process object detection results, build trajectories, or sequence control primitives. LMPs can be generated **hierarchically** by composing known functions (e.g., `get_obj_names()` using perception modules) or invoking other LMPs to define *undefined* functions:

```
# define function stack_objs_in_order(obj_names).
def stack_objs_in_order(obj_names):
    for i in range(len(obj_names) - 1):
        put_first_on_second(obj_names[i + 1], obj_names[i])
```

where `put_first_on_second` is an existing open vocabulary pick and place primitive (e.g., CLIPort [36]). For new embodiments, these active function calls can be replaced with available control APIs that represent the action space (e.g., `set_velocity`) of the agent. Hierarchical code-gen with verbose variable names can be viewed as a variant of chain of thought prompting [47] via functional programming. Functions defined by LMPs can progressively accumulate over time, where new LMPs can reference previously constructed functions to expand policy logic.

To execute an LMP, we first check that it is safe to run by ensuring there are no import statements, special variables that begin with `__`, or calls to `exec` and `eval`. Then, we call Python's `exec` function with the code as the input string and two dictionaries that form the scope of that code execution: (i) `globals`, containing all APIs that the generated code might call, and (ii) `locals`, an empty dictionary which will be populated with variables and new functions defined during `exec`. If the LMP is expected to return a value, we obtain it from `locals` after `exec` finishes.

### A. Prompting Language Model Programs

Prompts to generate LMPs contain two elements:
**1. Hints** e.g., import statements that inform the LLM which APIs are available and type hints on how to use those APIs.

```
import numpy as np
from utils import get_obj_names, put_first_on_second
```

**2. Examples** are instruction-to-code pairs that present few-shot "demonstrations" of how natural language instructions should be converted into code. These may include performing arithmetic, calling other APIs, and other features of the programming language. Instructions are written as comments directly preceding a block of corresponding solution code. We can maintain an LMP "session" by incrementally appending new instructions and responses to the prompt, allowing later instructions to refer back to previous instructions, like "undo the last action".

### B. Example Language Model Programs (Low-Level)

LMPs are perhaps best understood through examples, to which the following section builds up from simple pure-Python instructions to more complex ones that can complete robot tasks. These examples use OpenAI Codex `code-davinci-002` with `temperature` 0 (i.e., deterministic greedy token decoding). Here, the prompt (in gray) starts with a Hint to indicate we are writing Python. It then gives one Example to specify the format of the return values, to be assigned to a variable called `ret_val`. Input instructions are green, and generated outputs are highlighted:

```
# Python script
# get the variable a.
ret_val = a
# find the sum of variables a and b.
ret_val = a + b
# see if any number is divisible by 3 in a list called xs.
ret_val = any(x % 3 == 0 for x in xs)
```

**Third-party libraries.** Python code-writing LLMs store knowledge of many popular libraries. LMPs can be prompted to use these libraries to perform complex instructions without writing all of the code e.g., using NumPy to elicit spatial reasoning with coordinates. Hints here include import statements, and Examples define cardinal directions. Variable names are also important to indicate that `pts_np` and `pt_np` are NumPy arrays. Operations with 2D vectors imply that the points are also 2D. Example:

```
import numpy as np
# move all points in pts_np toward the right.
ret_val = pts_np + [0.3, 0]
# move a pt_np toward the top.
ret_val = pt_np + [0, 0.3]
# get the left most point in pts_np.
ret_val = pts_np[np.argmin(pts_np[:, 0]), :]
# get the center of pts_np.
ret_val = np.mean(pts_np, axis=0)
# the closest point in pts_np to pt_np.
ret_val = pts_np[np.argmin(np.sum((pts_np - pt_np)**2, axis=1))]
```

**First-party libraries.** LMPs can also use first-party libraries (perception or control primitive APIs) not found in the training data if those functions have meaningful names and are provided in Hints/Examples. For example (full prompt in B.2):

```
from utils import get_pos, put_first_on_second
...
# move the purple bowl toward the left.
target_pos = get_pos('purple bowl') + [-0.3, 0]
put_first_on_second('purple bowl', target_pos)
objs = ['blue bowl', 'red block', 'red bowl', 'blue block']
# move the red block a bit to the right.
target_pos = get_pos('red block') + [0.1, 0]
put_first_on_second('red block', target_pos)
# put the blue block on the bowl with the same color.
put_first_on_second('blue block', 'blue bowl')
```

The Hints import two functions for a robot domain: one to obtain
the 2D position of an object by name (using an open vocabulary
object detector [2]) and another to put the first object on the
second target, which can be an object name or a 2D position.
Note the LMP's ability to adapt to new instructions — the first
modifies the movement magnitude by using "a bit," while the
second associates the object with "the same color."

**Language reasoning** can be few-shot prompted using code-
writing LLMs (full prompt in B.1) to e.g., associate object
names with natural language descriptions ("sea-colored block"),
categories ("bowls"), or past context ("other block"):

```
objs = ['blue bowl', 'red block', 'red bowl', 'blue block']
# the bowls.
ret_val = ['blue bowl', 'red bowl']
# sea-colored block.
ret_val = 'blue block'
# the other block.
ret_val = 'red block'
```

### C. Example Language Model Programs (High-Level)

**Control flows.** Programming languages allow using control
structures such as if-else and loop statements. Previously
we showed LMPs can express for-loops in the form of list
comprehensions. Here we show how they can write a while-loop
can form a simple feedback policy. Note that the prompt (same
as the one in B.2) does not contain such Examples:

```
# while the red block is to the left of the blue bowl, move it to the
right 5cm at a time.
while get_pos('red block')[0] < get_pos('blue bowl')[0]:
    target_pos = get_pos('red block') + [0.05, 0]
    put_first_on_second('red block', target_pos)
```

**LMPs can be composed** via nested function calls. This allows
including more few-shot examples into individual prompts to
improve functional accuracy and scope, while remaining within
the LLM's maximum input token length. The following (full
prompt in B.4) generates a response that uses `parse_obj`, another
LMP that associates object names with language descriptions:

```
objs = ['red block', 'blue bowl', 'blue block', 'red bowl']
# while the left most block is the red block, move it toward the right.
block_name = parse_obj('the left most block')
while block_name == 'red block':
    target_pos = get_pos(block_name) + [0.3, 0]
    put_first_on_second(block_name, target_pos)
    block_name = parse_obj('the left most block')
```

The `parse_obj` LMP (full prompt in Appendix B.5):

```
objs = ['red block', 'blue bowl', 'blue block', 'red bowl']
# the left most block.
block_names = ['red block', 'blue block']
block_positions = np.array([get_pos(name) for name in block_names])
left_block_name = block_names[np.argmin(block_positions[:, 0])]
ret_val = left_block_name
```

**LMPs can hierarchically generate functions** for future reuse:

```
import numpy as np
from utils import get_obj_bbox_xyxy
# define function: total = get_total(xs).
def get_total(xs):
    return np.sum(xs)
# define function: get_objs_bigger_than_area_th(obj_names, bbox_area_th).
def get_objs_bigger_than_area_th(obj_names, bbox_area_th):
    return [name for name in obj_names
            if get_obj_bbox_area(name) > bbox_area_th]
```

Function generation can be implemented by parsing the code gen-
erated by an LMP, locating yet-to-be-defined functions, and calling
another LMP specialized in function-generation to create those
functions. This allows both the prompt and the code generated
by LMPs to call yet-to-be-defined functions. The prompt engineer
would no longer need to provide all implementation details in
Examples — a "rough sketch" of the code logic may suffice.
High-level LMPs can also follow good abstraction practices and
avoid "flattening" all the code logic onto one level. In addition
to making the resultant code easier to read, this improves code
generation performance as shown in Section IV-A. Locating yet-
to-be-defined functions is also done within the body of generated
functions. Note in the example above, `get_obj_bbox_area` is not
a provided API call. Instead, it can be generated as needed:

```
# define function: get_obj_bbox_area(obj_name).
def get_obj_bbox_area(obj_name):
    x1, y1, x2, y2 = get_obj_bbox_xyxy(obj_name)
    return (x2 - x1) * (y2 - y1)
```

Note the prompt did not specify exactly what `get_obj_bbox_xyxy`
returns, but the name suggests that it contains the minimum and
maximum xy coordinates of an axis-aligned bounding box, and
the LLM is able to infer this and generate the correct code.

In Python, we implement hierarchical function generation
by parsing a code block's abstract syntax tree and checking for
functions that do not exist in the given scope. We use the function-
generating LMP to write these undefined functions and add them
to the scope. This procedure is repeated on the generated function
body, hierarchically creating new functions in a depth-first manner.
**Combining control flows, LMP composition, and hierarchical
function generation.** The following example shows how LMPs
can combine these capabilities to follow more complex instructions
and perform a task in the tabletop manipulation domain. Prompts
are omitted for brevity, but they are similar to previous ones. The
high-level LMP generates high-level policy behavior and relies
on `parse_obj` to get object names by language descriptions:

```
objs = ['red block', 'blue bowl', 'blue block', 'red bowl']
# while there are blocks with area bigger than 0.2 that are left of the
red bowl, move them toward the right.
block_names = parse_obj('blocks with area bigger than 0.2 that are
                         left of the red bowl')
while len(block_names) > 0:
    for block_name in block_names:
        target_pos = get_pos(block_name) + np.array([0.1, 0])
        put_first_on_second(block_name, target_pos)
    block_names = parse_obj('blocks with area bigger than 0.2 that are
                            left of the red bowl')
```

Then, `parse_obj` uses `get_objs_bigger_than_area_th` (yet-to-
be-defined function), to complete the query. This function is given
through an import statement in the Hints of the `parse_obj` prompt,
but it is not implemented. Hierarchical function generation would
subsequently create this function as demonstrated above.

```
objs = ['red block', 'blue bowl', 'blue block', 'red bowl']
# blocks with area bigger than 0.2 that are left of the red bowl.
block_names = ['red block', 'blue block']
red_bowl_pos = get_pos('red bowl')
use_block_names = [name for name in block_names
                   if get_pos(name)[0] < red_bowl_pos[0]]
use_block_names = get_objs_bigger_than_area_th(use_block_names, 0.2)
ret_val = use_block_names
```

We describe more on prompt engineering in the Appendix A.

### D. Language Model Programs as Policies

In the context of robot policies, LMPs can compose perception-to-control feedback logic given natural language instructions, where the high-level outputs of perception model(s) (states) can be programmatically manipulated and used to inform the parameters of low-level control APIs (actions). Prior information about available perception and control APIs can be guided through Examples and Hints. These APIs "ground" the LMPs to a real-world robot system, and improvements in perception and control algorithms can directly lead to improved capabilities of LMP-based policies. For example, in real-world experiments below, we use recently developed open-vocabulary object detection models like ViLD [3] and MDETR [2] off-the-shelf to obtain object positions and bounding boxes.

The benefits of LMP-based policies are threefold: they (i) can adapt policy code and parameters to new tasks and behaviors specified by unseen natural language instructions, (ii) can generalize to new objects and environments by bootstrapping off of open-vocabulary perception systems and/or saliency models, and (iii) do not require any additional data collection or model training. The generated plans and policies are also interpretable as they are represented in code, allowing for easy modification and reuse. Using LMPs for high-level user interactions inherits the benefits of LLMs, including parsing expressive natural language with commonsense knowledge, taking prior context into account, multilingual capabilities, and engaging in dialog. In the experiment section that follows, we demonstrate multiple instantiations of LMPs across different robots and different tasks, showcasing the approach's flexible capabilities and ease of use.

## IV. EXPERIMENTS

The goals of our experiments are threefold: (i) evaluate the impact of using hierarchical code generation (across different language models) and analyze modes of generalization, (ii) compare Code as Policies (CaP) against baselines in simulated language-instructed manipulation tasks, and (iii) demonstrate CaP on different robot systems to show its flexibility and ease-of-use. Additional experiments can be found in the Appendix, such as generating reactive controllers to balance a cartpole and perform end-effector impedance control (Appendix F). The Appendix also contains the prompt and responses for all expriments. Full videos and Colab Notebooks that reproduce these experiments can be found on the website.

### A. Hierarchical LMPs on Code-Generation Benchmarks

We evaluate our code-generation approach on two code-generation benchmarks: (i) a robotics-themed RoboCodeGen and (ii) HumanEval [1], which consists of standard code-gen problems.

TABLE I: Hierarchical code-generation solves more problems in RoboCodeGen (in % pass rates) and improves with scaling laws (as # model parameters increases).

| Method | GPT-3 [12] | | Codex [1] | |
|---|---|---|---|---|
| | 6.7B | 175B | cushman | davinci |
| Flat | 3 | 68 | 54 | 81 |
| Hierarchical | **5** | **84** | **57** | **95** |

TABLE II: Hierarchical code-gen performs better (% pass rate) on generic coding problems from HumanEval [1]. Greedy is decoding LLM with temperature=0. P@N evaluates correctness across N samples with temperature=0.8.

| | Greedy | P@1 | P@10 | P@100 |
|---|---|---|---|---|
| Flat | 45.7 | 34.9 | 75.1 | 90.9 |
| Hierarchical | **53.0** | **39.8** | **80.6** | **95.7** |

**RoboCodeGen**: we introduce a new benchmark with 37 function generation problems with several key differences from previous code-gen benchmarks: (i) it is robotics-themed with questions on spatial reasoning (e.g., find the closest point to a set of points), geometric reasoning (e.g., check if one bounding box is contained in another), and controls (e.g., PD control), (ii) using third-party libraries (e.g. NumPy) are both allowed and encouraged, (iii) provided function headers have no docstrings nor explicit type hints, so LLMs need to infer and use common conventions, and (iv) using not-yet-defined functions are also allowed, which can be created with hierarchical code-gen. Example benchmark questions can be found in Appendix E. We evaluate on four LLMs accessible from the OpenAI API[1]. As with standard benchmarks [1], our evaluation metric is the percentage of the generated code that passes human-written unit tests.

See Table I. Domain-specific language models (Codex models), generally perform better, and within each model family, performance improves with larger models. Hierarchical performs better across the board, showing the benefit of allowing the LLM to break down complex functions into hierarchical parts and generate code for each part separately.

We also analyze how code generation performance varies across the five types of generalization proposed in [23]. Hierarchical helps Productivity the most, which is when the new instruction requires longer code, or code with more logic layers than those in Examples. These improvements however, only occur with the two davinci models, and not cushman, suggesting that a certain level of code-generation capability needs to be reached first before hierarchical code-gen can bring further improvements. More results are in Appendix E.2.

Evaluations in **HumanEval** [1] demonstrate that hierarchical code-gen improves not only policy code, but also general-purpose code. See Table II. Numbers achieved are higher than in recent works [1], [11], [58]. Note that we use `code-davinci-002`, while previous works use `code-davinci-001`, but the relative improvements with hierarchical are consistent across the board. More details in Appendix D.

### B. CaP: Drawing Shapes via Generated Waypoints

In this domain, we task a real UR5e robot arm to draw various shapes by generating and following a series of 2D waypoints. For

---

[1] Two text models: the 6.7B GPT-3 model [12] and 175B InstructGPT [22]. Two Codex models [1]: cushman and davinci, trained to generate code. davinci is larger and better. Sizes of Codex models are not public.

TABLE III: Success rates over task families with 50 episodes per task.

| Train/Test | Task Family | CLIPort [36] | NL Planner | CaP (ours) |
|---|---|---|---|---|
| SA SI | Long-Horizon | 78.80 | 86.40 | **97.20** |
| SA SI | Spatial-Geometric | **97.33** | N/A | 89.30 |
| UA SI | Long-Horizon | 36.80 | 88.00 | **97.60** |
| UA SI | Spatial-Geometric | 0.00 | N/A | **73.33** |
| UA UI | Long-Horizon | 0.00 | 64.00 | **80.00** |
| UA UI | Spatial-Geometric | 0.01 | N/A | **62.00** |

perception, the LMPs are given APIs that detect object positions, implemented with off-the-shelf open vocabulary object detector MDETR [2]. For actions, an end-effector trajectory following API is provided. There are four LMPs: (i) parse user commands, maintain a session, and call action APIs, (ii) parse object names from language descriptions, (iii) parse waypoints from language descriptions, and (iv) generate new functions. Examples of successful on-robot executions of unseen language commands are in Fig. 2c. The system can elicit spatial reasoning to draw entirely new shapes from language commands. Additional examples which demonstrate the ability to parse precise dimensions, manipulate previous shapes, and multi-step commands, as well as full prompts, are in Appendix H.

### C. CaP: Pick & Place Policies for Table-Top Manipulation

The table-top manipulation domain tasks a UR5e robot arm to pick and place various plastic toy objects on a table. The arm is equipped with a suction gripper and an in-hand Intel Realsense D435 camera. We provide perception APIs that detect the presences of objects, their positions, and bounding boxes, via MDETR [2]. We also provide a scripted primitive that picks an object and places it on a target position. Prompts are similar to those from the last domain, except trajectory parsing is replaced with position parsing. Examples of on-robot executions of unseen language commands are in Fig. 2 panels a and b, showing the capacity to reason about object descriptions and spatial relationships. Other commands that use historical context (e.g., "undo that"), reason about objects via geometric (e.g., "smallest") and spatial (e.g., "right-most") descriptions are in Appendix I.

### D. CaP: Table-Top Manipulation Simulation Evaluations

We evaluate CaP on a simulated table-top manipulation environment from [16], [18]. The setup tasks a UR5e arm and Robotiq 2F85 gripper to manipulate 10 colored blocks and 10 colored bowls. We inherit all 8 tasks, which are referred as "long-horizon" tasks because of their multi-step nature (e.g., "put the blocks in matching bowls"). We additionally define 6 tasks to evaluate the spatial-geometric reasoning capabilities of CaP (e.g., "place the blocks in a diagonal line"). Each task is parameterized by some attributes (e.g., "pick up <obj> and place it in <corner>"). We split the task instructions (I) and the attributes (A) into "seen" (SI, SA) and "unseen" categories (UI, UA), where "seen" means it's allowed to appear in the prompts or be trained on (in the case of supervised baseline). More details in Appendix K. We consider two baselines: (i) language-conditioned multi-task CLIPort [36] policies trained via imitation learning on 30k demonstrations, and (ii) few-shot prompted LLM planner using natural language instead of code.

Results are in Table III. CaP compares competitively to the supervised CLIPort baseline on tasks with seen attributes and instructions, despite only few-shot prompted with one example rollout for each task. With unseen task attributes, CLIPort's performance degrades significantly, while LLM-based methods retain similar performance. On unseen tasks and attributes, end-to-end systems like CLIPort struggle to generalize, and CaP outperforms LLM reasoning directly with language (also observed in [20]). Moreover, the natural-language planners [14], [16]–[18] are not applicable for tasks that require precise numerical spatial-geometric reasoning. We additionally show the benefits reasoning with code over natural language (both direct question and answering and Chain of Thought [47]), specifically the ability of the former to perform precise numerical computations, in Appendix C.

### E. CaP: Mobile Robot Navigation and Manipulation

In this domain, a robot with a mobile base and a 7 DoF arm is tasked to perform navigation and manipulation tasks in real-world kitchen. For perception, the LMPs are given object detection APIs implemented via ViLD [3]. For actions, the robot is given APIs to navigate to locations and grasp objects via both names and coordinates. Examples of on-robot executions of unseen language commands are in Fig. 2. This domain shows that CaP can be deployed across realistic tasks on different robot systems with different APIs. It also illustrates the ability to follow long-horizon reactive commands with control structures as well as precise spatial reasoning, which cannot be easily accomplished by prior works [16], [17], [36]. See prompts and additional examples in Appendix J.

## V. DISCUSSION AND LIMITATIONS

CaP generalizes at a specific layer in the robot stack: interpreting natural language instructions, processing perception outputs, then parameterizing low-dimensional inputs to control primitives. This fits into systems with factorized perception and control, and it imparts a degree of generalization (acquired from pretrained LLMs) without the magnitude of data collection needed for end-to-end learning. Our method also inherits LLM capabilities unrelated to code writing e.g., supporting instructions with non-English languages or emojis (Appendix L. CaP can also express *cross-embodied* plans that perform the same task differently depending on the available APIs (Appendix M). However, this ability is brittle with existing LLMs, and it may require larger ones trained on domain-specific code.

CaP today are restricted by the scope of (i) what the perception APIs can describe (e.g., no visual-language models to date can describe whether a trajectory is "bumpy" or "more C-shaped"), and (ii) which control primitives are available. Only a handful of named primitive parameters can be adjusted without over-saturating the prompts. CaP also struggle to interpret commands that are significantly longer or more complex, or operate at a different abstraction level than the given Examples. In the tabletop domain, it would be difficult for LMPs to "build a house with the blocks," since there are no Examples on building complex 3D structures. Our approach also assumes all given instructions are feasible, and we cannot tell if a response will be correct a priori.

### ACKNOWLEDGEMENTS

Special thanks to Vikas Sindhwani, Vincent Vanhoucke for helpful feedback on writing, Chad Boodoo for operations and hardware support.

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

### A. Prompt Engineering

Using LMPs to reliably complete tasks via code generation requires careful prompt engineering. While these prompts do not have to be long, they do need to be relevant and specific. Here, we discuss a few general guidelines that we followed while developing prompts for this paper.

It is very important for prompts to contain code that *has no bugs*. Bugs in the prompt lead to unreliable and incorrect responses. Conversely, if the LMP is writing incorrect code for a given Instruction, the prompt engineer should first verify that the prompt, especially the Examples most closely related to the Instruction, is bug-free. To reduce bugs related to syntax errors, one simple method is writing prompts in a code editor with syntax highlighting.

There are many cases where the prompt contains variables or functions whose names are ambiguous. To produce reliable responses under these conditions, Examples in the prompt should treat these ambiguities consistently. If a variable named `point` is treated as an `numpy.ndarray` object in one Example and as a `shapely.geometry.Point` object in another, the LMP will not be able to "decide" on which convention to use, resulting in unreliable responses. Another way to handle ambiguity is by providing informal type hints, such as appending `_np` to variable names to indicate its type, or appending it to function names to indicate the type of the returned variable. In general, more specific variable and function names give more consistent results.

For using third party libraries, including import statements in the prompt may not be necessary, as we found that LMPs can generate code that calls NumPy and SciPy without them. However, explicit import statements do improve reliability and increase the chance of LMPs using those libraries when the need arises. For using first party libraries, meaningful function names that follow popular conventions (e.g., begin with `set_` and `get_`) and specify return object formats (e.g., `get_bbox_xyxy`) induce more accurate usages. Import statements in the Hints should be formatted as if we were importing functions. For example, in Python this means using `from utils import function_name` instead of `import function_name`. If the latter is used, the LMP may treat the imported name as a package, and the generated code might write `function_name.function_name()`.

One type of LMP failures are related to code generation correctness. For example, minor coding mistakes when calling internal or external APIs, such as missing arguments, can be fixed with an Hint or Example demonstrating the correct usage. Incorrect assumptions on variable types can also be fixed in similar fashions. Other coding failures may be addressed by descriptive function names to encourage appropriate library usage (`perform_function_with_np()`) or succinct code logic (`# implement in one line.`) While it is possible to use LLMs to edit code and fix bugs (e.g., by using OpenAI's code edit API), in our experience this yielded inconsistent results (not always able to correct mistakes, and sometimes changed what the function was doing), so we did not employ this method in our experiments.

### B. Method Section Prompts

*1) Language-based reasoning:* Full prompt:

```
objs = ['green block', 'green bowl', 'yellow block', 'yellow bowl']
# the yellow block.
ret_val = 'yellow block'
# the blocks.
ret_val = ['green block', 'yellow block']
```

*2) First-party:* Full prompt:

```
from utils import get_pos, put_first_on_second
objs = ['gray block', 'gray bowl']
# put the gray block on the gray bowl.
put_first_on_second('gray block', 'gray bowl')
objs = ['purple block', 'purple bowl']
# move the purple bowl toward the left.
target_pos = get_pos('purple bowl') + [-0.3, 0]
put_first_on_second('purple bowl', target_pos)
```

*3) Combining language reasoning, third-party, and first-party libraries.:* Full prompt:

```
import numpy as np
from utils import get_pos, put_first_on_second
objs = ['cyan block', 'cyan bowl', 'pink bowl']
# put the cyan block in cyan bowl.
put_first_on_second('cyan block', 'cyan bowl')
objs = ['gray block', 'silver block', 'gray bowl']
# place the top most block on the gray bowl.
names = ['gray block', 'silver block']
positions = np.array([get_pos(name) for name in names])
name = names[np.argmax(positions[:,1])]
put_first_on_second(name, 'gray bowl')
objs = ['purple block', 'purple bowl']
# put the purple bowl to the left of the purple block.
target_pos = get_pos('purple block') + [-0.3, 0]
put_first_on_second('purple bowl', target_pos)
```

*4) LMPs can be composed.:* Full prompt:

```
import numpy as np
from utils import get_pos, put_first_on_second, parse_obj
objs = ['yellow block', 'yellow bowl', 'gray block', 'gray bowl']
# move the sun colored block toward the left.
block_name = parse_obj('sun colored block')
target_pos = get_pos(block_name) + [-0.3, 0]
put_first_on_second(block_name, target_pos)
objs = ['white block', 'white bowl', 'yellow block', 'yellow bowl']
# place the block closest to the blue bowl on the other bowl.
block_name = parse_obj('the block closest to the blue bowl')
bowl_name = parse_obj('a bowl other than the blue bowl')
put_first_on_second(block_name, bowl_name)
```

*5) parse_obj prompt.:* Full prompt:

```
import numpy as np
from utils import get_pos
objs = ['brown bowl', 'green block', 'brown block', 'green bowl']
# the blocks.
ret_val = ['brown block', 'green block']
# the sky colored block.
ret_val = 'blue block'
objs = ['orange block', 'cyan block', 'purple bowl', 'gray bowl']
# the right most block.
block_names = ['orange block', 'cyan block']
block_positions = np.array([
            get_pos(block_name) for block_name in block_names])
right_block_name = block_names[np.argmax(block_positions[:, 0])]
ret_val = right_block_name
```

### C. Reasoning with Code vs. Natural Language

To investigate how robot-relevant reasoning through LLMs can be performed with LMPs rather than with natural language, we created a benchmark that consists of two sets of tasks: (i) selecting objects in a scene from spatial-geometric descriptions, and (ii) selecting position coordinates from spatial-geometric descriptions. Object selection has 28 questions with commands such as "find the name of the block closest to the blue bowl," where a list of block and bowl positions are provided as input context in the prompt. Position selection has 23 questions with commands such as

"interpolate 3 points on a line from the cyan bowl to the blue bowl." An LLM-generated answer for position selection is considered correct if all coordinates are within 1cm of the ground truth.

We evaluate LMPs against two variants of reasoning with natural language: (i) **Vanilla**, given a description of the setting (e.g., list of object positions) and the question, directly outputs the answer (e.g., "Q: What is the top-most block?" → "A: red block"), and (ii) **Chain of Thought (CoT)** [47], which performs step-by-step reasoning given examples of intermediate steps in the prompt (e.g., encouraging the LLM to list out y-coordinates of all blocks in the scene before identifying the top-most block).

TABLE IV: Using code for spatial-geometric reasoning yields higher success rate (mean %) than using vanilla natural language or chain-of-thought prompting.

| | Natural Language | | Code |
|---|---|---|---|
| Tasks | Vanilla | CoT [47] | LMP (ours) |
| Object Selection | 39 | 68 | **96** |
| Position Selection | 30 | 48 | **100** |
| Total | 35 | 58 | **98** |

Results in Table IV show that LMPs achieve accuracies in the high 90s, outperforming CoT, which outperforms Vanilla. CoT enables LLMs to reason about relations and orders (e.g. which coordinate is to the right of another coordinate), but failures occur for precise and multi-step numerical computations. By contrast, code from LMPs can use Python to perform such computations, and they often leverage external libraries to perform more complex operations (e.g., NumPy for vector addition). CoT and LMPs are not mutually exclusive – it is possible to prompt "step-by-step" code-generation to solve more complex tasks via CoT, but this is a direction not explored in this work.

### D. CodeGen HumanEval Additional Results

Here we provide additional results to our HumanEval experiments. In total, three variants of the bigger Codex model (code-davinci-002) are tested. Our approach is Hier. CodeGen + Hier Prompts, where the prompt encourages the LLM to call yet-to-be-defined functions by including such examples. For comparisons, we evaluate against Flat CodeGen + No Prompt, essentially just using the LLM directly, and Flat CodeGen + Flat Prompt, for fair comparison with flat code-generation, since our hierarchical approach has a prompt. The prompts only contain only 2 Examples:

**Prompt for Flat CodeGen:**

```
prompt_f_gen_flat = '''
def get_total(xs: List[float]) -> float:
    """Find the sum of a list of numbers called xs.
    """
    return sum(xs)
# end of function

def get_abs_diff_between_means(xs0: List[float],
                               xs1: List[float]) -> float:
    """Get the absolute difference between the means of two
    lists of numbers.
    m0 = sum(xs0) / len(xs0)
    m1 = sum(xs1) / len(xs1)
    return abs(m0 - m1) # end of function
```

**Prompt for Hierarchical CodeGen:**

```
def get_total(xs: List[float]) -> float:
    """Find the sum of a list of numbers called xs.
    """
    return sum(xs)
# end of function

def get_abs_diff_between_means(xs0: List[float],
                               xs1: List[float]) -> float:
    """Get the absolute difference between the means of two lists of
    numbers.
    """
    m0 = get_mean(xs0)
    m1 = get_mean(xs1)
    return abs(m0 - m1)
# end of function
```

Note the only difference in the hierarchical prompt is using a yet-to-be-defined function `get_mean` instead of calculating the mean directly. This "allows" the LLM to generate code that also call yet-to-be-defined functions.

We report pass rates for when using the most likely outputs ("greedy", which is done by setting temperature to 0), as well as pass rates for at least one solution from sampling various numbers of solutions (1, 10, and 100) with temperature set to 0.8, similar to those used in prior works [1], [11], [58].

TABLE V: Hierarchical code generation also performs better (in % pass rates) on generic coding problems from the standard HumanEval benchmark [1]. For columns, Greedy means decoding LLM with temperature=0, while P@N means evaluating correctness across N samples decoded from LLM with temperature=0.8.

| | Greedy | P@1 | P@10 | P@100 |
|---|---|---|---|---|
| code-davinci-001 [11] | - | 36.0 | - | 81.7 |
| PaLM Coder [11] | - | 36.0 | - | 88.4 |
| Flat CodeGen + No Prompt | 45.7 | 34.9 | 75.1 | 90.9 |
| Flat CodeGen + Flat Prompts | 50.6 | 36.6 | 77.6 | 93.3 |
| Hier. CodeGen + Hier Prompts | **53.0** | **39.8** | **80.6** | **95.7** |

See results in Table II. In all instances hierarchical code generation outperforms flat code generation, and the numbers achieved are higher than those reported in recent works [1], [11], [58] (although previous works evaluated against code-davinci-001, which is likely not as performant as code-davinci-002). Out of the 164 questions in HumanEval, 6.5% led to hierarchical code generation, but of which both Flat CodeGen variants got 44% success, while Hier CodeGen code got 56%. While success rate when sampling 100 responses is above 90% across the board, we note that sampling multiple solutions is not practical for LMPs, which need to perform tasks in a zero-shot manner without engineering prior unit tests. As such, for LMPs we always set temperature to 0 and use the most likely output.

### E. Robot Code-Generation Benchmark

*1) Example Questions:* Here are four types of benchmark questions and their examples:

- Vector operations with Numpy:
  ```
  pts = interpolate_pts_np(start, end, n)
  ```
- Simple controls:
  ```
  u = pd_control(x_curr, x_goal, x_dot, Kp, Kv)
  ```
- Manipulating shapes with shapely:
  ```
  circle = make_circle(radius, center)
  ```
- Using first-party libraries:
  ```
  ret_val = obj_shape_does_not_contain_others(obj_name,
  other_obj_names)
  ```

For the last type, we provide imports of first-party functions, like ones that get object geometric information by name, as Hints in the prompt.

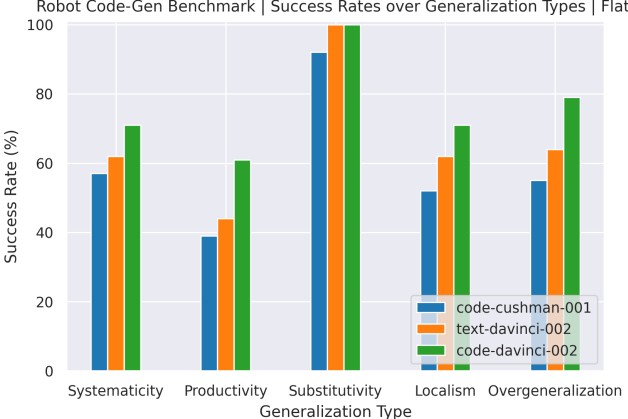

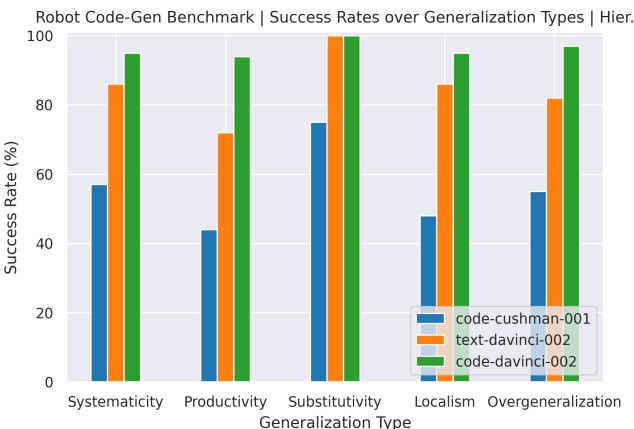

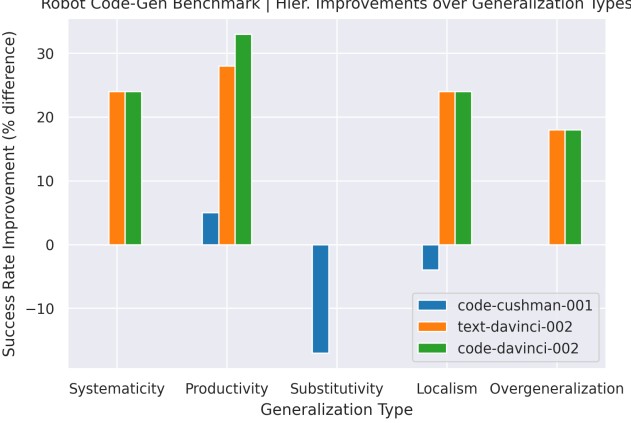

Fig. 4: Robot Code-Generation Benchmark Performance across Generalization Types for Flat (top) and Hierarchical (middle) Code-Generation, as well as the performance improvements made by Hierarchical Code-Generation (bottom).

*2) Generalization Analysis:* We analyze how well code-generation performs across the fives types of generalizations described in [23], where generalization is evaluated by comparing the examples given in the prompt with the new instructions given in the benchmark. We give a description of the five types of generalization applied to our benchmark. Specifically, we say that solving a problem in the benchmark demonstrates a particular type of generalization if the problem's instruction or solution satisfy the following conditions:

- Systematicity: recompose parts of Examples' instructions or code snippets.
- Productivity: have longer code or contains more levels (e.g., hierarchical function calls) than Examples.
- Substitutivity: use synonyms or replace words of similar categories from Examples.
- Localism: reuse seen parts or concepts for different purposes.
- Overgeneralization: use new API calls or programming language features not seen in Examples.

In Figure 4 we report results across three models. The top plot shows using flat prompts and flat code generation, while the bottom uses hierarchical prompts and hierarchical code generation. The relative rankings of the three models are consistent across all generalization types, with code-davinci-002 performing the best. The smallest model, code-cushman-001, performs especially poorly in Productivity and Localism, but it performs much better on Substitutivity. Indeed, the high Substitutivity performance across all models may indicate that language models are particularly robust to replacing words that have similar meanings or are from similar categories, while generalization to more complex commands (longer answers via Productivity or maintaining local structures via Localism) are more challenging. Hierarchical code generation also brings significant performance improvements for both davinci models, especially on Productivity, allowing these models to write longer and more multi-layered functions. However, these improvements are not apparent for cushman, which actually has lower Substitutivity and Localism success rates, suggesting reaching a certain code-generation capability level is needed before Hierarchical Code-Generation can bring further improvements.

### F. CaP: Reactive Controllers for Toy Tasks

LMPs can express reactive controllers, and we demonstrate this on: (i) OpenAI Gym's CartPole balancing and (ii) end-effector position impedance control. Using the same prompt structure, both controllers can be generated zero-shot, reflecting the knowledge already stored in the language model, which can be invoked when the objective is sufficiently specified in the function signature:

```
# define function: direction = keep_pole_upright_with_pd_control(x,
x_dot, theta, theta_dot).
# info: direction is 1 if going right, 0 if going left.
def keep_pole_upright_with_pd_control(x, x_dot, theta, theta_dot):
    # define constants.
    kp = 1
    kd = 1
    # define direction.
    direction = 1
    # define error.
    error = theta
    # define error_dot.
    error_dot = theta_dot
    # define control.
    control = kp * error + kd * error_dot
    # define direction.
    if control < 0:
        direction = 0
    return direction
```

The generated LMP expresses a controller that takes as input the position of the cart, angle of the pole, and their velocities, then it balances the pole by actuating the cart below with discrete left or right actions. This controller is only a simple one that requires initialization to be near upright, but it can stabilize an unstable fixed point.

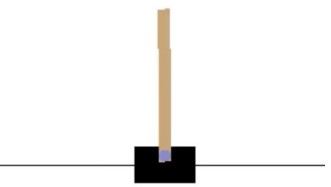

Fig. 5: LMPs can balance a cartpole

LMPs can likewise be prompted to express impedance control:

```
# define function: tau = ee_impedance_control(x_curr, x_goal,
                          x_dot, K_x_mat, D_x_mat, J).
def ee_impedance_control(x_curr, x_goal, x_dot, K_x_mat,
                         D_x_mat, J):
    x_err = x_goal - x_curr
    x_dot_err = -x_dot
    tau = np.matmul(J.T,
          np.matmul(K_x_mat, x_err) + np.matmul(D_x_mat, x_dot_err))
    return tau
```

to move a robot arm end-effector towards a goal position with joint torques. The controller is functional in that it can control a UR5e robot in PyBullet, but simplified in that it does not compensate for Coriolis or gravity forces. Note the need to include extra information about the expected direction output as well as the hint to use PD control in the function signature. Without these hints, the resultant function may still look reasonable (e.g. it may output continuous values for control instead of discrete), but it will not work for this specific environment API. For the names of the input gains, `_mat` was needed for the LMP to treat them as matrices instead of scalars, and `_x` was needed to indicate these gains were for the end-effector, not the joints. We demonstrate the use of this controller by commanding the end-effector 3D positions of a UR5e robot in PyBullet. The default PD gains of 1 also work in this domain without additional tuning as the CartPole environment is relatively simple. More complex continuous control tasks may require actually tuning the gains based on execution feedback, something our method does not support at the moment.

Both examples show it is possible to generate simple reactive controllers, but more work is needed to express more complex ones.

### G. Visual Language Models

For real-world experiments, we use off-the-shelf open-vocabulary object detection models, ViLD [3] and MDETR [2] to perform object detection, localization, and segmentation. These are called visual language models because they take as input a natural language description (caption) of the image and try to find objects in that description. ViLD is used for the mobile robot domain, while MDETR is used for the tabletop manipulation and whiteboard drawing domains. Both models give an axis-aligned bounding box in the image along with per-pixel segmentation masks of the detected objects. To convert these detections to 3D coordinates for both perception and action (e.g., scripted picking primitives), we deproject the corresponding pixels from a depth camera, whose transform to the robot frame is registered

a priori. The robustness of today's vision language models could still be improved, and many real-world failures could be attributed to inaccurate detections. In addition, a degree of prompt engineering is also required for VLMs. For example, MDETR detects blocks more reliably with the word "square" than "block," and applying our approach to a new domain will require some prompt engineering for the vision language model.

### H. Whiteboard Drawing

In this domain, a UR5e robot is tasked to draw and erase various shapes described by natural language on a whiteboard. A dry-erase marker is rigidly attached to the robot end-effector. The whiteboard dimensions, location, and the location of the eraser are known. Additional objects may be added to the scene for the commands to refer to (e.g., draw a circle around the blue block). In our demos, we use Google Cloud's speech-to-text and text-to-speech APIs to allow users interact with the system through voice commands and also hear the robot's responses to commands.

**Prompts.**
- `draw_ui`: the high-level
  UI for parsing user commands and calling other functions
  https://code-as-policies.github.io/prompts/draw_ui.txt
- `parse_obj_name`:
  return names of objects from natural language descriptions
  https://code-as-policies.github.io/prompts/parse_obj_name.txt
- `parse_shape_pts`: return sequence of
  2D waypoints of shapes from natural language descriptions
  https://code-as-policies.github.io/prompts/parse_shape_pts.txt
- `transform_shape_pts`: performs 2D transforms on
  a sequence of 2D points from natural language descriptions
  https://code-as-policies.github.io/prompts/transform_shape_pts.txt
- `function_generation`: define functions from comments
  https://code-as-policies.github.io/prompts/fgen_simple.txt

**APIs.**
- `get_obj_names()` - gets list of available objects in the scene. these are prespecified.
- `get_obj_pos(name)` - get the 2D position of the center of an object by name.
- `draw(pts_2d)` - draws a shape by commanding the robot end-effector to follow a squence of points on the whiteboard. The robot first moves to a point above the first point in the trajectory, moves down to until contact with the whiteboard is detected, and proceeds to follow the rest of the trajectory.
- `erase(pts_2d)` - erases a shape by commanding the robot end-effector to first establish contact with a eraser (eraser position is hardcoded) before following the the rest of the trajectory.

**Instructions.** These instructions were given to the robot in series from an initial blank whiteboard. See full video and generated code on the website.

1) draw a 5cm hexagon around the middle
2) draw a line that bisects the hexagon
3) make them both bigger

4) erase the hexagon and the line
5) draw the sun as a circle at the top right
6) draw the ground as a line at the bottom
7) draw a pyramid as a triangle on the ground
8) draw a smaller pyramid a little bit to the left
9) draw circles around the blocks
10) draw a square around the sweeter fruit

### I. Real-World Tabletop Manipulation

In this domain, a UR5e robot is tasked to manipulate objects on a tabletop according to natural language instructions. The robot is equipped with a suction gripper, and it can only perform pick and place actions parameterized by 2D top-down pick and place positions. The robot is also expected to answer questions about the scene (e.g., how many blocks are there?) by using the provided perception APIs. In our demos, we use Google Cloud's speech-to-text and text-to-speech APIs to allow users interact with the system through voice commands and also hear the robot's responses to commands and questions.

**Prompts.**

- `tabletop_ui`: the high-level
  UI for parsing user commands and calling other functions
  https://code-as-policies.github.io/prompts/tabletop_ui.txt
- `parse_obj_name`:
  return names of objects from natural language descriptions
  https://code-as-policies.github.io/prompts/parse_obj_name.txt
- `parse_position`:
  return a 2D position from natural language descriptions
  https://code-as-policies.github.io/prompts/parse_position.txt
- `parse_question`: return a response (could be a number, a boolean, or a string) to a natural language question
  https://code-as-policies.github.io/prompts/parse_question.txt
- `function_generation`: define functions from comments
  https://code-as-policies.github.io/prompts/fgen.txt

**APIs.**

- `get_obj_names()` - gets list of available objects in the scene. these are prespecified.
- `get_obj_pos(name)` - gets the 2D position of the center of an object by name.
- `is_obj_visible(name)` - checks if an object is visible by name.
- `get_bbox(name)` - gets the 2D axis-aligned bounding box of an object by name. This is in robot base coordinates, not in pixels.
- `get_segmask(name)` - gets the segmentation mask of an object detection by name. This is in pixels.
- `get_color_rgb(name)` - gets the average RGB color of an object detection crop by name.
- `get_corner_name(pos_2d)` - gets the name of the corner (e.g., top right corner) closest to the 2d point.
- `get_side_name(pos_2d)` - gets the name of the side (e.g., left side) closest to the 2d point.
- `denormalize_xy(normalized_pos_2d)` - converts a normalized 2D coordinate (each value between 0 and 1) to an actual 2D coordinate in robot frame.
- `put_first_on_second(obj_name, target)` - picks the first object by name and places it on top of the target by name. The target could be another object name or a 2D position. Picking and placing are done by moving the suction gripper directly on top of the desired positions, moving down until contact is detected, then either engages or disengages the suction cup.
- `say(message)` - uses the robot speaker to voice out a message.

We demonstrate CaP on three domains in the tabletop manipulation setting. Instructions of each domain are listed below and were performed in a sequence. See full videos and generated code on the website.

**Instructions for 4 blocks domain.**

1) Put the blocks in a horizontal line near the top
2) Move the sky-colored block in between the red block and the second block from the left
3) Why did you move the green block?
4) Which block did you move?
5) Arrange the blocks in a square around the middle
6) Make the square bigger
7) Undo that
8) rotate the square by 45 degrees
9) Can you throw blocks?
10) Move the red block 5cm to the bottom
11) Do the same with the other blocks
12) Put the blocks on different corners clockwise starting at the top right corner

**Instructions for 3 blocks and 3 bowls domain.**

1) Put the red block to the left of the rightmost bowl
2) Now move it to the side farthest away from it
3) How many bowls are to the left of the red block?
4) place the blocks in bowls with non matching colors
5) put the blocks in a vertical line 20 cm long and 10 cm below the blue bowl
6) imagine that the bowls represent a volcano, a forest, and an ocean
7) also imagine that the blocks are parts of a building
8) now build a tower in the forest
9) show me what happens when a volcano erupts over the ocean

**Instructions for fruits, bottles, and plates domain.**

1) How many fruits are there?
2) Tell me their names
3) Are there any fruits on the green plate?
4) Move all fruits to the green plate and bottles to the blue plate
5) Move the smallest fruit back to the yellow plate
6) Wait until you see an egg and put it on the green plate
7) Put the darkest object in the plate that has the apple

### J. Mobile Robot

The mobile manipulation experiment is set up with a Everyday Robots robot navigating and interacting with objects in a real world office kitchen. The robot has a mobile base and a 7DoF arm. For implementing the perception APIs, we mainly use the RGBD camera sensor on the robot. The robot is shown in Fig. 6.

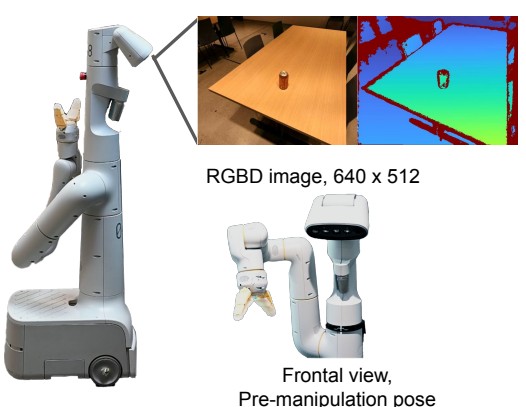

RGBD image, 640 x 512

Frontal view,
Pre-manipulation pose

Fig. 6: Experiment Setup for mobile manipulation with a Everyday Robots robot.

**Prompts.**
- `mobile_ui`: the high-level
  UI for parsing user commands and calling other functions
  https://code-as-policies.github.io/prompts/mobile_ui.txt
- `parse_obj_name`:
  return names of objects from natural language descriptions
  https://code-as-policies.github.io/prompts/mobile_parse_obj_name.txt
- `parse_position`:
  return a 2D position from natural language descriptions
  https://code-as-policies.github.io/prompts/mobile_parse_pos.txt
- `transform_traj`: performs 2D transforms on
  a sequence of 2D points from natural language descriptions
  https://code-as-policies.github.io/prompts/mobile_transform_traj.txt
- `function_generation`: define functions from comments
  https://code-as-policies.github.io/prompts/fgen_simple.txt

**APIs.**
- `get_obj_names()` - gets list of available objects in the scene. these are prespecified.
- `get_obj_pos(name)` - get the 2D position of the center of an object by name.
- `is_obj_visible(name)` - returns whether or not the robot sees an object by name.
- `get_visible_obj_names()` - returns a list of currently visible object names.
- `get_loc_names()` - returns a list of all predefined location names the robot can navigate to.
- `get_obj_pos(name)` - gets the 3D location of an object by name. This object must be currently visible.
- `get_loc_pos(name)` - gets the 2D location and 1D angle of a predefined location.
- `get_robot_pos_and_angle` - gets the current 3D robot position and 1D angle (heading).
- `goto_pos(pos_3d)` - navigates to a 3D position by running the robot's internal motion planner.
- `goto_loc(name)` - navigates to a location by name by running the robot's internal motion planner.
- `pick_obj(name)` - picks up an object by its name. The object must be currently visible. This is implemented as a scripted picking primitive using ViLD object detections.

- `place_at_pos(pos_3d)` - places the currently held object at a position.
- `place_at_obj(name)` - places the currently held object on top of another object by name.
- `say(message)` - uses the robot's speaker to voice out a message.

Below we list commands that were performed on the mobile robot platform. The first are navgiation-related tasks, while the second are manipulation related. For the latter manipulation commands, note the ability of CaP to form "short-term memory" by explicitly record variables (in this case, the robot's past positions) in the Python execution scope and referring back them later. See videos and generated code on the website.

**Mobile Navigation Instructions.**
1) Moving in a 3m by 2m rectangle around the office chair
2) Do that again but rotated 45 degrees clockwise
3) Go in a 1.5m square around the barstool as many times as needed, check each step if there is a banana, only stop moving when you see the banana
4) Follow the convex hull containing the chairs
5) Move back and forth between the table and the countertop 3 times

**Mobile Manipulation Instructions.**
1) How many snacks are on the table?
2) Take the water bottle from the desk and put it in the middle of the fruits on the table
3) This is the compost bin
4) This is the recycle bin
5) This is the landfill bin
6) The coke can and the apple are on the table
7) Put way the coke can and the apple on their corresponding bins

*K. Simulation Tabletop Manipulation Evaluations*

Similar to the real-world tabletop domain, we construct a simulated tabletop environment, in which a UR5e robot equipped with a Robotiq 2F85 jaw gripper is given natural language instructions to complete rearrangement tasks. The objects include 10 different colored blocks and 10 different colored bowls. The proposed CaP is given APIs for accessing a list of present objects and their locations, via a scripted object detector, as well as a pick-and-place motion primitive that are parameterized by either coordinates or object names.

**Prompts.**
- `tabletop_ui`: the high-level
  UI for parsing user commands and calling other functions
  https://code-as-policies.github.io/prompts/sim_tabletop_ui.txt
- `parse_obj_name`:
  return names of objects from natural language descriptions
  https://code-as-policies.github.io/prompts/sim_parse_obj_name.txt
- `parse_position`:
  return a 2D position from natural language descriptions
  https://code-as-policies.github.io/prompts/sim_parse_position.txt

```python
import numpy as np
from utils import get_pos, put_first_on_second
objects = ['cyan block', 'cyan bowl', 'pink bowl']
# put the cyan block in cyan bowl.
put_first_on_second('cyan block', 'cyan bowl')
objects = ['gray block', 'silver block', 'gray bowl']
# place the top most block on the gray bowl.
names = ['gray block', 'silver block']
positions = np.array([get_pos(name) for name in names])
name = names[np.argmax(positions[:,1])]
put_first_on_second(name, 'gray bowl')
objects = ['purple block', 'purple bowl']
# put the purple bowl to the left of the purple block.
target_pos = get_pos('purple block') + [-0.3, 0]
put_first_on_second('purple bowl', target_pos)
objects = ['red block', 'red bowl', 'blue block', 'blue bowl']
# 把蓝色的木块放在红色的碗里.
put_first_on_second('blue block', 'red bowl')
objects = ['green block', 'cyan block']
# move the 🟩 10cm to the 👈.
target_pos = get_pos('green block') + [-0.1, 0]
put_first_on_second('green block, target_pos)
```

Fig. 7: LMPs inherit benefits of LLMs, such as parsing commands from non-English languages and emojis.

- `function_generation`: define functions from comments
  https://code-as-policies.github.io/prompts/fgen.txt

**APIs.**

- `get_obj_names()` - gets list of available objects in the scene. these are prespecified.
- `get_obj_pos(name)` - gets the 2D position of the center of an object by name.
- `denormalize_xy(normalized_pos_2d)` - converts a normalized 2D coordinate (each value between 0 and 1) to an actual 2D coordinate in robot frame.
- `put_first_on_second(obj_name, target)` - picks the first object by name and places it on top of the target by name. The target could be another object name or a 2D position. Picking and placing are done by moving the suction gripper directly on top of the desired positions, moving down until contact is detected, then either engages or disengages the suction cup.

We evaluate CaP and the baselines on the following tasks, where each task refers to a unique instruction template (e.g., "Pick up the <block> and place it in the corner <distance> to the <bowl>") that are parameterized by certain attributes (e.g., <block>). We split the tasks into the instructions and the attributes to "seen" and "unseen" categories, where the "seen" instructions or attributes are permitted to appear in the prompt or used for training (in the case of supervised baselines). Full list can be found below. Note that we further group the instructions into "Long-Horizon" and "Spatial-Geometric" task families. The "Long-Horizon" instructions are 1-5 in Seen Instructions and 1-3 in Unseen Instructions. The "Spatial-Geometric" instructions are 5-8 in Seen Instructions and 4-6 in Unseen Instructions.

**Seen Instructions.**

1) Pick up the <block1> and place it on the (<block2> or <bowl>)
2) Stack all the blocks
3) Put all the blocks on the <corner/side>
4) Put the blocks in the <bowl>
5) Put all the blocks in the bowls with matching colors
6) Pick up the block to the <direction> of the <bowl> and place it on the <corner/side>
7) Pick up the block <distance> to the <bowl> and place it on the <corner/side>
8) Pick up the <nth> block from the <direction> and place it on the <corner/side>

**Unseen Instructions.**

1) Put all the blocks in different corners
2) Put the blocks in the bowls with mismatched colors
3) Stack all the blocks on the <corner/side>
4) Pick up the <block1> and place it <magnitude> to the <direction> of the <bowl>
5) Pick up the <block1> and place it in the corner <distance> to the <bowl>
6) Put all the blocks in a <line> line

**Seen Attributes.**

1) <block>: blue block, red block, green block, orange block, yellow block
2) <bowl>: blue bowl, red bowl, green bowl, orange bowl, yellow bowl
3) <corner/side>: left side, top left corner, top side, top right corner
4) <direction>: top, left
5) <distance>: closest
6) <magnititude>: a little
7) <nth>: first, second
8) <line>: horizontal, vertical

**Unseen Attributes.**

1) <block>: pink block, cyan block, brown block, gray block, purple block
2) <bowl>: pink bowl, cyan bowl, brown bowl, gray bowl, purple bowl
3) <corner/side>: bottom right corner, bottom side, bottom left corner
4) <direction>: bottom, right
5) <distance>: farthest
6) <magnititude>: a lot
7) <nth>: third, fourth
8) <line>: diagonal

In Table VI we provide detailed simulation results that report task success rats for fine-grained task categories. Attributes refer to <> fields, the values of which can be seen by the method (e.g., training set for CLIPort, prompt for language-based methods). Instructions refer to the templated instruction type given in each row, which can also be seen or unseen. A total of 50 trials are performed per task, each with sampled attributes and initial scene configurations (block and bowl types, numbers, and positions). Note that CLIPort by itself (no oracle) is just a feedback policy and it does not know when to stop — in this case we run 10 actions from the CLIPort policy and evaluate success at the end. To improve CLIPort performance, we use a variant that uses oracle information from the simulation to stop the policy when success is detected (oracle termination).

*L. Additional LLM Capabilities*

Because we use code-davinci-002, a code-writing model fine-tuned from the more general purpose davinci GPT-3, our

TABLE VI: Detailed simulation tabletop manipulation success rate (%) across different task scenarios.

| | CLIPort (oracle termination) | CLIPort (no oracle) | NL Planner | CaP (ours) |
|---|---|---|---|---|
| **Seen Attributes, Seen Instructions** | | | | |
| Pick up the <object1> and place it on the (<object2> or <recepticle-bowl>) | 88 | 44 | 98 | **100** |
| Stack all the blocks | **98** | 4 | 94 | 94 |
| Put all the blocks on the <corner/side> | **96** | 8 | 46 | 92 |
| Put the blocks in the <recepticle-bowl> | **100** | 22 | 94 | **100** |
| Put all the blocks in the bowls with matching colors | 12 | 14 | **100** | **100** |
| Pick up the block to the <direction> of the <recepticle-bowl> and place it on the <corner/side> | **100** | 80 | N/A | 72 |
| Pick up the block <distance> to the <recepticle-bowl> and place it on the <corner/side> | 92 | 54 | N/A | **98** |
| Pick up the <nth> block from the <direction> and place it on the <corner/side> | **100** | 38 | N/A | 98 |
| Total | 85.8 | 33.0 | 86.4 | **94.3** |
| Long-Horizon Total | 78.8 | 18.4 | 86.4 | **97.2** |
| Spatial-Geometric Total | **97.3** | 57.3 | N/A | 89.3 |
| **Unseen Attributes, Seen Instructions** | | | | |
| Pick up the <object1> and place it on the (<object2> or <recepticle-bowl>) | 12 | 10 | 98 | **100** |
| Stack all the blocks | 96 | 8 | 96 | **100** |
| Put all the blocks on the <corner/side> | 0 | 0 | 58 | **100** |
| Put the blocks in the <recepticle-bowl> | 46 | 0 | 88 | **96** |
| Put all the blocks in the bowls with matching colors | 30 | 26 | **100** | 92 |
| Pick up the block to the <direction> of the <recepticle-bowl> and place it on the <corner/side> | 0 | 0 | N/A | **60** |
| Pick up the block <distance> to the <recepticle-bowl> and place it on the <corner/side> | 0 | 0 | N/A | **100** |
| Pick up the <nth> block from the <direction> and place it on the <corner/side> | 0 | 0 | N/A | **60** |
| Total | 23.0 | 5.5 | 88.0 | **88.5** |
| Long-Horizon Total | 36.8 | 8.8 | 88.0 | **97.6** |
| Spatial-Geometric total | 0.0 | 0.0 | N/A | **73.3** |
| **Unseen Attributes, Unseen Instructions** | | | | |
| Put all the blocks in different corners | 0 | 0 | 60 | **98** |
| Put the blocks in the bowls with mismatched colors | 0 | 0 | **92** | 60 |
| Stack all the blocks on the <corner/side> | 0 | 0 | 40 | **82** |
| Pick up the <object1> and place it <magnitude> to the <direction> of the <recepticle-bowl> | 0 | 0 | N/A | **38** |
| Pick up the <object1> and place it in the corner <distance> to the <recepticle-bowl> | 4 | 0 | N/A | **58** |
| Put all the blocks in a <line> | 0 | 0 | N/A | **90** |
| Total | 0.7 | 0.0 | 64.0 | **71.0** |
| Long-Horizon Total | 0.0 | 0.0 | 64.0 | **80.0** |
| Spatial-Geometric Total | 1.3 | 0.0 | N/A | **62.0** |

method inherits capabilities of LLMs that a pure code-writing model may not have. For example, we can leverage the LLM's generality to parse commands from non-English languages as well as emojis. See Figure 7.

### M. Cross Embodiment Example

CaP exhibit a degree of cross-embodiment support [59], [60] by performing the same task differently depending on the action APIs. In the example below, we give Hints of the action APIs, and the resultant plan changes depending on the whether or not the robot is omnidirectional or unidirectional. We note that this ability is brittle with existing LLMs and cannot reliably adapt to APIs

that are very different. More robustness may require larger ones trained on domain-specific code.

```
# omnidirectional robot.
# available actions: move_up(dist), move_right(dist), move_back(dist).
# make a U shape 5 meters wide and 10 meters long.
robot.move_back(dist=10)
robot.move_right(dist=5)
robot.move_up(dist=10)
# unidirectional robot.
# available actions: turn_left(deg), move_forward(dist).
# make a U shape 3 meters wide and 8 meters long.
robot.move_forward(dist=8)
robot.turn_left(deg=90)
robot.move_forward(dist=3)
robot.turn_left(deg=90)
robot.move_forward(dist=8)
```