# OpenReview forum: "Code as Policies: Language Model Programs for Embodied Control"
_robot-learning.org/CoRL/2022/Workshop/LangRob — LangRob 2022 Poster_

### Official Review · Reviewer_AmD3 · 2022-11-12
**Good paper, compelling results.**

**Rating:** 9
**Confidence:** 4

**Review:**

This paper presents CaP, a framework for prompting LLMs to generate robot-executable code. The key idea is to compose perception and control APIs by showing a pre-trained LLM some examples of desired programmatic behaviors. CaP achieves compelling real-world and simulation results by solving unseen variations of tasks, e.g., generating a program to draw hexagons (a new shape) by decomposing the task into draw line commands.

Strengths:
+ Impressive real-world results that showcase generalization to unseen task variations.
+ Benchmark and Colab notebooks are publically available.
+ Code generation does not require additional training or fine-tuning, just prompting(!)
+ Detailed descriptions of prompts used.

Weaknesses:
- It’s unclear how robust LLM based code-generation is. How often does the generated code throw runtime exceptions? If an error occurs, how is the user expected to correct it?
- While prompting is cool, it still requires someone to provide additional context regarding the scene and robot. Like the coordinate system used to command velocities, and the range of physical parameters for low-level primitives. As such, is an expected user for CaP, someone who has a good understanding of both programming robots and also wants to control the robot casually through natural language commands? Or a robot expert that helps in setting up a system for casual users?

---

### Official Review · Reviewer_qchw · 2022-11-13
**LLMs for writing robotics policies**

**Rating:** 8
**Confidence:** 4

**Review:**

This paper presents Code as Policies (CaP): a framework for leveraging code-writing LLMs to generate policies for robots that connect to existing perception and control modules. This paper introduces a method for hierarchical code-gen where functions can be defined and then later written by the LLM to generate complex behaviors. The paper performs extensive evaluations on the impact of how scaling impacts the LLMs ability to generate policies and how it generalizes on a wide suite of tasks.

Overall, I recommend this paper be accepted. Extracting policies rather than just plans from LLM is very interesting, and the potential for using LLM for policy generation is well demonstrated in the paper. Below are my suggestions for the paper:

- The authors should consider looking at the contemporary work ProgPrompt [1], which also proposes using LLMs to generate code structures for robotic behavior.
- I am somewhat skeptical of the practical use of CaP for generating low-level controllers, especially when parameter tuning is important. As the authors note in the cartpole example, the choice of hyperparameters for PID control worked out since the task is relatively simple when starting near the fixed point, but for a complex domain the challenge is in tuning these parameters. It would be interesting if CaP could actually generate useful parameters for variable impedance control across different robotic tasks (e.g: door opening, table wiping, etc.). However, the application for high-level task planning is very promising.
- I am curious about how sensitive this approach is to the importance of related prompts for getting good results; an experimental validation would be useful for this.
- How does CaPs perform when there are multiple object instances (e.g: two blue blocks, or two tables) ? It seems like in most experiments, there are just single objects of each instance (blocks of different colors, a single table, etc.)
- I'm curious how CaPs would do for complex spatial reasoning tasks where the geometries of the environment are relevant for task planning. For example, when navigating a building, the robot may need to open doors, or to pick objects up, it may need to open drawers to access them. I wonder how CaPs would handle task and motion planning problems that involve multiple modalities [2]
- In the Related Work, there is a missing “)” in the third sentence


[1] Singh, Ishika, et al. "Progprompt: Generating situated robot task plans using large language models." arXiv preprint arXiv:2209.11302 (2022).
[2] Garrett, Caelan Reed, et al. "Integrated task and motion planning." Annual review of control, robotics, and autonomous systems 4 (2021): 265-293.

---

### Decision · Program_Chairs · 2022-11-15

Accept (Poster)